# Investigating Fertility Intentions for a Second Child in Contemporary China Based on User-Generated Content

**DOI:** 10.3390/ijerph17113905

**Published:** 2020-05-31

**Authors:** Ying Qian, Xiao-ying Liu, Bing Fang, Fan Zhang, Rui Gao

**Affiliations:** Department of Information Management, School of Management, Shanghai University, Shanghai 200444, China; qian.ying@t.shu.edu.cn (Y.Q.); vickyliu0923@shu.edu.cn (X.-y.L.); zfanele@i.shu.edu.cn (F.Z.); ruigao0901@i.shu.edu.cn (R.G.)

**Keywords:** fertility intention, second child, machine–human hybrid approach, user-generated content, two-child policy

## Abstract

China’s two-child policy, aimed at boosting the country’s total fertility rate, has failed to achieve the desired outcomes. Previous studies on low fertility rates mainly used data obtained from demographic censuses, questionnaires, or interviews. These data-gathering methods are costly, entailing time delays and yielding limited information. User-generated content (UGC) provides an alternative data source. We propose a machine–human hybrid approach using UGC obtained from social media to assess users’ intentions to have a second child. Our results showed that couples associate a second child with high economic costs mainly through negative impacts on the mothers’ careers, with no concomitant economic benefits. A key motivation for having two children relates to the mental benefit of the joy in having children. However, raising a second child also entails considerable mental costs such as exhaustion and pressure. Couples largely seek help within their extended families, that is, their parents are major sources of child-rearing support. Therefore, the government should devise ways of reducing the negative impacts of having a second child on a woman’s career and provide child-rearing support to help increase the fertility rate. Our proposed approach can also be used to elicit the reasons for low fertility rates in other countries.

## 1. Introduction

China launched its two-child policy in 2015, with the aim of countering the declining trend in the fertility rate and reducing the pace at which Chinese society is aging [1]. According to United Nations statistics, the proportion of the elderly (aged 60 years and above) within China’s total population is expected to more than double from 12.4% (168 million) in 2010 to 25% (402 million) in 2030 (see Figure 1). According to this estimation, the proportion of the elderly could exceed 30% by 2040, placing considerable pressure on governmental budgets for supporting the growing elderly population.

China’s total fertility rate has remained below the replacement rate for over a decade (see Figure 2). The two-child policy was intended to increase the total fertility rate. However, the fertility rate continues to hover at extremely low levels, as youngest couples are not willing to have two children [2]. In China, most couples have one child as a responsibility to keep the kinship. Therefore, changes in the total fertility rate are highly contingent on families’ decisions on whether to have a second child.

As the two-child policy is unique to China and has only been in force since 2015, there are few existing studies on the fertility rate relating to the fertility intentions for a second child. Studies on the fertility rate have mostly considered the total fertility rate, focusing on issues such as the low fertility rate [3,4] and the fertility transition [5,6]. Several macro-level factors influencing low fertility have been identified, such as industrialization and urbanization [7,8,9], the importance of education [10], the decline in infant and child mortality [11], and women economic independence [12]. 

Urbanization improved living conditions, leading to a decline in infant and child mortality [11]. The drop in infant mortality led to the reduction in replacement fertility [13], the adapted fertility behavior [14], and the increased incentive for investments in each child [10]. Meanwhile, new technologies rapidly emerged with industrialization. The demand for human capital correspondingly increased, requiring parents to invest more in their children’s education. Rather than having more children, parents preferred to have children who were educated. Thus, a decline in fertility was evident [15,16,17,18,19]. With further societal development, women became better educated and more independent. Highly educated women tend to have fewer children as marriage and childbearing were delayed to complete their study [20]. Moreover, working women usually restrict the number of children they have to avoid the loss of career opportunities [12,21,22,23,24]. Thus, women’s educational levels and employment statuses are important factors contributing to the decline in fertility rates [12,25].

In China, childbearing attitudes and behaviors are rooted in Confucian philosophy and traditions of ancestor worship [26,27]. Since the 1970s, the Chinese government implemented a series of birth control policies with the concern of the population growing out of control, and the fertility rate dropped significantly in the following years [28,29,30]. In current China, smaller families are preferred [31]. The macro-level conditions offer limited insights to formulate a countermeasure as they do not provide detailed information on individual families’ needs regarding child-rearing. We systematically investigated the micro-level factors influencing the willingness of Chinese couples to have two children, applying user-generated content (UGC).

UGC has emerged as a promising source of data for ascertaining individuals’ actual thoughts and opinions [32]. UGC is easy to obtain, with detailed information and can be efficiently processed at a low cost using advanced natural language processing (NLP) and machine learning technology. As human language cannot be fully grasped by existing technologies, a hybrid approach that combines machines and humans is normally applied. Machine learning can efficiently extract keywords and sentences while human interventions enable further analysis of their meanings. 

This study had two primary goals. First, we aimed to introduce a methodological framework, entailing a machine–human approach for analyzing UGC to better understand the support required for child-rearing. Second, we applied the proposed framework to develop a holistic picture of the factors affecting Chinese couples’ decisions on whether to have a second child based on UGC from social networks. This framework could be applied across different countries and regions. 

## 2. Study Design 

We developed a machine–human hybrid approach to mine individuals’ feelings about having a second child from UGC sourced from the Internet. The development of natural language processing (NLP) and machine learning technology has enabled the automatic extraction from UGC of critical information that is repeatedly discussed by users. However, semantic content cannot be directly understood by existing NLP computer programs. Therefore, we conducted a further manual analysis to judge the keywords, which are those related to the fertility intention of a second child, and to categorize them with reference to the corresponding sentences. The results obtained using this hybrid approach are more accurate than those obtained using NLP technology and machine learning alone. Moreover, the analysis is more effective than one conducted solely by human researchers. The proposed approach comprised five stages, as illustrated in Figure 3.

According to the framework, the study entailed five stages that are described below. 

Stage 1: Data Collection

We collected UGC data from “Zhihu”, the largest Chinese social question answering platform. Zhihu, which started in December 2010, is a social question answering platform targeting better educated customers, providing long detailed answers discussing various questions raised also by users. Based on a report in 2017, the number of registered users had reached 160 million [33] and news in April 2019 reported that Zhihu had attracted more than 220 million users [34]. With the fast growth of its users group, the Zhihu platform is becoming more representative of the opinions of the general public [33]. Zhihu is a platform where people can learn from each other and gain better understanding of the world though raising questions and providing answers openly and freely. Users also can upvote for answers expressing their support for the answer. The data from Zhihu is open to the general public, and can be used in opinion mining [35,36].

Responses to the two questions “Why Chinese government’s policies encouraging a second child have limited effect?” and “Do you feel happy about having two children?” were crawled. The first question related to the discussion of policies. However, an inspection of the answers showed that most people explained the reasons why they did not want to have a second child, which led to the limited effect of the second-child policy. For those irrelevant answers, we eliminated the keywords in step 5. For the second question, most people talked about their own experience with a second child. Some talked about the life of their friends or relatives with a second child, which were also personal experiences related to our study. There were 478 responses (a total of 70,109 words, upvoted by 20,255 users) to the first question and 527 responses (a total of 181,718 words, upvoted by 14,590 users) to the second question.

Stage 2: Preprocessing of the Raw UGC Data

As the raw UGC data were written in Chinese, it was necessary to segment the sentences into words. We availed Jieba, a Python software for Chinese text segmentation to segment the raw UGC data.

The next step entailed the removal of stop words. In the raw UGC data, there were many stop words (e.g., “the” or “and”) that significantly affected the efficiency of NLP and needed to be removed. Table 1 presents a comparison of the preprocessed and processed data. The examples for preprocessing of the raw data are shown in Table 1.

Stage 3: Training Word Embeddings

Word embeddings are processes that map words on to a numerical vector space, enabling the semantic similarity between words to be calculated. We utilized the skip-gram model to train the word embeddings, as it takes contextual information into consideration in the word embedding training. Skip-gram is a type of artificial neural network model that transforms the inputted words into embedded word vectors within the hidden output layer. This model was implemented via the Word2Vec model in Python. The training results were saved as word vectors for use during the subsequent stages.

Stage 4: Clustering Keywords

The same semantics can be expressed in several ways. The repetitiveness of the UGC will reduce the efficiency of the manual review. To avoid repetitiveness, we applied the machine learning approach to cluster the keywords according to the embedded word vectors. We applied the “k-means” principle to cluster the keywords. First, the distance between words was measured. Second, we randomly designated words to serve as the cluster center for each category. Third, words are categorized within the nearest cluster. The fourth step entailed recalculating the cluster centers. Iteration of the third and fourth steps continues until there are no further changes in the cluster center. Finally, we selected the representative word in each cluster. After identifying the keywords, we deployed NLP technology to extract the corresponding sentences of the keywords from the UGC. Thus, we were able to elucidate the keywords’ semantics during the next stage.

Stage 5: Manual Extraction of the Details of Users’ Opinions

As existing NLP technologies are not capable of directly understanding textual semantics, we have to judge whether the keywords are related to fertility intention manually and then categorize the keywords with reference to the corresponding sentences. According to Easterlin and Crimmins, the demand for children is related the relative price (raising cost) of children. Economic factors (both benefits and costs) are related to the fertility intention [37]. From our clustered keywords, we also identified mental factors, including mental benefits and mental costs, related to the fertility intention. Therefore, we categorized these keywords within the following groups: economic benefits, economic costs, mental benefits, mental costs, and external factors.

Five graduate students participated in this exercise. Initially, they categorized the keywords independently. Subsequently, they cross-examined each other, and eventually, through discussion, reached a consensus, achieving consistent results.

## 3. Results

Below, we elucidate the in-depth reasons underlying Chinese couples’ decisions on whether or not to have a second child based on an analysis of the five categories of keywords and corresponding sentences. A holistic view with detailed information emerged from the data, as shown in Figure 4.

The keywords in blue are related to the economic aspect, those in pink are related to the mental aspect, and those in brown are related to external factors. The size and position of the keywords represent their relative importance: keywords that are bigger and listed near the center are more important.

### 3.1. Economic Benefits

Two keywords relating to economic benefits were identified: “a prosperous and happy life” and “old age security”, see Table 2. Chinese society was formerly based on agriculture. More children within families meant a larger work force for farming. Moreover, the parents were assured of a prosperous and happy life when they were old as their children would provide old age security. As a result, for a long period of Chinese history, having more children was associated with economic benefits. However, the sentences corresponding to these keywords convey an opposing view. Most contemporary Chinese parents do not expect their children to provide them with economic support. Some couples mentioned that improved social security corresponds to a reduced desire to have children, indicating that old age security is no longer associated with having more children and is instead associated with social security. Thus, there are few economic benefits linked to having children in the current era.

### 3.2. Economic Costs

Many keywords emerged for economic costs that were categorized under three subcategories, see Table 3. The first was short-term expenditure associated with child-rearing for meeting the daily life requirements of a child, such as “diapers”, “feed”, and “money”. A second subcategory was long-term expenditure on, for example, “housing” and “extra-curricular classes”, which require major investments in the child’s long-term future. The third subcategory was indirect expenditure related to the negative impacts of having a second child on a woman’s career development, as reflected in keywords such as “resign” and “career”, which represent the loss of future income.

The economic cost that most people mentioned was “resign” (8.4%), indicating that the topmost concern related to having a second child is its negative impact on women’s careers. It is hard for women to work and simultaneously take care of two children. Having one child, most mothers are able to continue their work with the support of their parents or parents-in-law or a temporary maid. However, having a second child, most mothers will need to resign from their jobs to take care of the kids, at least for several years. It is a big negative impact on mother’s career development.

“Housing” ranked second (5.5%) among economic costs. Having a second child could entail having to stay with parents or parents-in-law so that the grandparents can help to look after the two children. A lack of sufficient space for a relatively large family (six or seven members comprising grandparents, parents, and children) would lead to friction. Further, couples choose apartments based on their locations in relation to kindergartens, primary schools, and junior high middle schools where their children could be admitted. To gain access to better educational facilities, couples may consider buying or changing into better-located apartments. However, the prices of apartments that are in proximity to good educational facilities are considerably higher than those of other apartments. Chinese couples are keen to provide their children with a good education. Therefore, the issue of long-term investments is also a key concern.

By contrast, short-term expenditure was not a major concern. Thus, while keywords such as “diaper,” “feed,” and “raise” were mentioned, they were not emphasized.

### 3.3. Mental Benefits

Mental benefits featured prominently in the UGC data, see Table 4. Many users expressed their feelings of happiness, pointing to the joy that their children had brought to their lives. These benefits are reflected in keywords such as “kiss” and “intimate” used in sentences such as “As soon as I arrive home from work, my son will cling to me. It is our intimate time.” In some cases, the children have been easy to raise and have not created any trouble for their families. They are very sweet and bring joy to the whole family. Users mentioned keywords such as “angel” in their comments: “Both of my children are angels. They hardly cry and just groan a little when they feel sick or uncomfortable.” Notably, many users stated that they do not regret having a second child because of these mental benefits. This view is indicated by keywords such as “amiable” used in the following sentence: “I am very happy to see that my two children are so amiable and considerate and that they are very close to each other. Isn’t this the original motivation for having a second child?”.

Among all the keywords identified, “Kiss” was mentioned by most users (16.3%), indicating that the associated mental benefit is the key motivation for having a second child.

### 3.4. Mental Costs

The number of users who mentioned mental costs exceeded that who mentioned the economic costs, which were categorized under three subcategories, see Table 5. The first was the exhaustion entailed in bringing up children that included “lactation period”, “sickness”, and “care and attention”. A second subcategory related to keywords conveyed tremendous pressure on parents, illustrated by “bear” and “collapse”. A lack of free time to “dining out” and “shopping” also relates to mental pressure. The third subcategory concerns the relationship between couples, as reflected in the keyword “Husband and wife relationship”. Many couples mentioned that having a second child has a negative impact on the husband and wife relationship.

The most-mentioned mental cost was exhaustion from raising a child, especially during the lactation period and when children are sick. Raising children undoubtedly requires tremendous efforts on the part of the parents. However, we found it surprising that many users mentioned that having a second child had a negative impact on the relationship between the husband and the wife.

### 3.5. External Factors

Table 6 shows that the most commonly mentioned external factors influencing the decision to have a second child were close family members, including parents-in-law, parents, the first child, and family and friends. In many cases, adhering to old Chinese traditions, parents or parents-in-law are in favor of couples having more children, sometimes urging them to have a second child. Some close family members were also willing to help raise the second child. We identified this factor as an important one that influenced the decision on whether to have a second child. Without the help provided by parents or parents-in-law, the experience of raising two children would be difficult. One mother wrote about her difficult experience: “When I had the second child, the first one is less than 3 years old. There was no one to help me to take care of the children. When I went out, I held the first child by the hand and the second child in my arms. I was so tired.”

These keywords reveal that parents and parents-in-law importantly influence the decision on whether to have a second child. If they can help to take care of the children, couples are more willing to have a second child as their support would help to alleviate the strain for them. Without their help, most couples are apprehensive about coping with these difficulties on their own. Evidently, society has done little to provide support that would make it easier for couples to have a second child.

Based on the above results, we found that the keywords mentioned for mental benefits and mental costs are more in number than the keywords mentioned for economic benefits and cost, as shown in Figure 5, which illustrated that mental benefits and costs are important factors in people’s decisions whether to have a second child.

## 4. Discussion

Using a machine–human hybrid methodology, we developed a comprehensive understanding of fertility intentions for a second child. We found that the factors influencing people’s decisions of whether to have a second child identified from this approach are closely related to the social economic condition of concurrent Chinese society.

Most previous studies about fertility preferences focused mainly on the economic costs of childbearing [2,3,4]. However, in fact, we found that the most important factors influencing people’s decisions of whether to have a second child in current China are related to mental aspects: the mental costs prevent people from having a second child, while mental benefits are the most prominent reasons for people to have a second child. Economic cost also has played a role but economic benefit is the least prominent factor influencing people’s decisions. The reasons for this are analyzed below:

Having more children today generates little economic benefits but incurs high economic costs. In traditional Chinese culture, children provided economic support for their elderly parents. However, in China, a modern pension system has been established since 1997, and the number of people covered under the national pension insurance program exceeds 900 million now [38]. Elderly people, especially urban-based seniors, are now living on their own pensions. Therefore, there is no economic motivation for having more children, which is a major reason explaining the low intention to have more than one child.

Raising children in China today associates with high economic costs, which has a strongly negative impact on the fertility rate. According to data obtained from the Chinese Academy of Social Sciences, in 2018, the total cost of raising a child up to the age of 16 years in China was 317,000 RMB on average [39]. Moreover, apart from being associated with monetary expenditure, economic costs extend to the impeding of women’s careers. According to the findings of an economic study conducted by the Royal Bank of Canada, “in the year following the birth of a first child, women aged 25 to 34 saw their earnings fall by almost half compared to women with no children,” and “women experience a significant earnings penalty spanning five years after the birth of a child.” [40].

The motivation for having more children mainly derives from the joy associated with children, which relates to the mental benefits. However, the mental costs for raising children are also enormous.

Well-natured children are a source of happiness not only to parents but also to other family members. An only child may grow up lonely. Siblings are good companions, which provide more social interactions and opportunities to learn from one another. Seeing two children getting along well is of great accomplishment [41].

However, raising children is not always enjoyable. Couples may also encounter mental costs such as disrupting the stability and harmony of conjugal relations, depression, and stress. Research showed that in China, as high as 37.2% of new mothers could suffer from postpartum depression [42]. Previous studies of child-rearing costs have taken little account of mental stress. Moreover, when designing policies to encourage couples to have a second child, the Chinese government has mainly focused on reducing economic costs, with little consideration given to alleviating the mental pressure entailed in raising children.

Last but not least, external factors play an important role in the fertility intention for a second child. Our findings indicated that in contemporary China, couples usually take family members’ advice into consideration when deciding whether or not to have a second child. This is because couples in China often rely on their parents’ support in raising the second child. In 2018, 6.97 million children were left behind by their parents in rural areas; of these children, 96% were being taken care of by their grandparents [43]. Couples who lack the support of their parents are thus less likely to have a second child.

In light of our findings, several strategic directions emerged for increasing the fertility rate in relation to having more than one child in China.

First, the negative impact of having a second child on mothers’ career development is a serious concern. Strategies for limiting such negative impacts are therefore necessary. The provision of support for raising children, especially for couples with children under the age of 3 years should be considered so that they still have time to focus on their work. Suggested policy recommendations include permitting couples to take their children to work while they are being breastfed and establishing state-owned nurseries for children between the ages of 1 and 3 years.

Second, efforts should be made to reduce the mental pressure experienced by couples with young children. For example, communities could organize activities for parents to share their child-raising experiences and help and support each other. Studies have shown that such activities are an effective means of releasing mental pressure.

Third, the provision of professional guidance and training could help couples to raise their children more easily and efficiently. Scientific child-rearing methods are conducive to children’s health and can greatly reduce the cost of raising children.

## 5. Conclusions

In conclusion, through combining natural language processing and machine learning to mine UGC on social media, we developed a hybrid machine–human approach to investigate fertility intentions, which is applicable not only in China but also in other countries and regions. We also developed a comprehensive picture for understanding the reasons for low fertility intentions for a second child in current China based on an analysis of UGC. In light of our findings, we have proposed practical policy recommendations to increase couples’ willingness to have a second child.

Our research has limitations that call for future research. Firstly, the main limitation of this research lies in the fact that our conclusions are based on the answers to the two questions from a social question answering platform. The UGC data came from one single source and the amount of the data is limited. In the future, we can utilize UGC data from various sources such as Weibo (microblogging), Wechat, and danmaku, where people discuss their opinions over related videos. A diversified data source could cover more perspectives from various user groups. Secondly, when we collect UGC data from social networks, we can also collect users’ demographic information, such as age, gender, socioeconomic status, and marital and family status, among others. In this way, we can further analyze and compare the fertility intentions of different population groups. Thirdly, we must improve the efficiency of the approach. The key sentences identified by the extracted keywords can also be clustered into groups by a machine learning method. In this way, to investigate the meaning of what people have written, we do not need to read each sentence but just one sentence from cluster, which can reduce the workload of manual analysis and further improve the efficiency of the approach. Lastly, even though more and more people are using the Internet, it is still impossible to cover everyone. The approach we proposed can be used as a method to conduct research together with other traditional social science methods. This approach is an efficient means that can get the opinions of many people in a short time.

## Figures and Tables

**Figure 1 ijerph-17-03905-f001:**
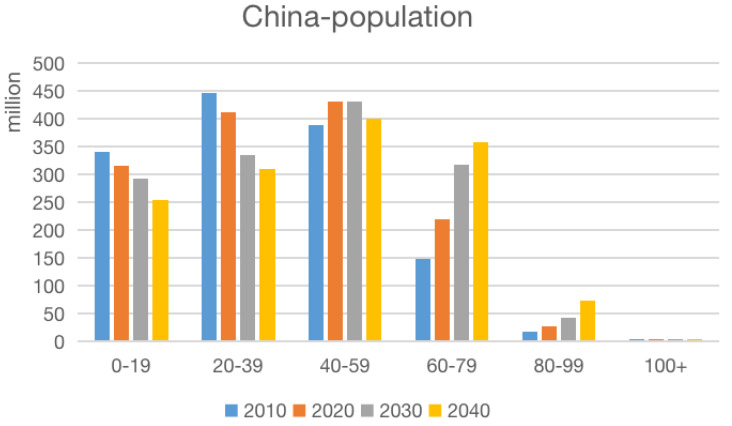
Estimated future age cohorts within China’s population.

**Figure 2 ijerph-17-03905-f002:**
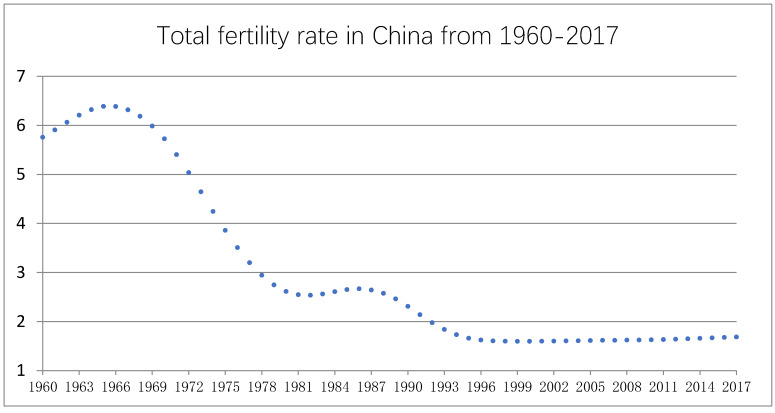
Total fertility rates in China for the period 1960–2017.

**Figure 3 ijerph-17-03905-f003:**
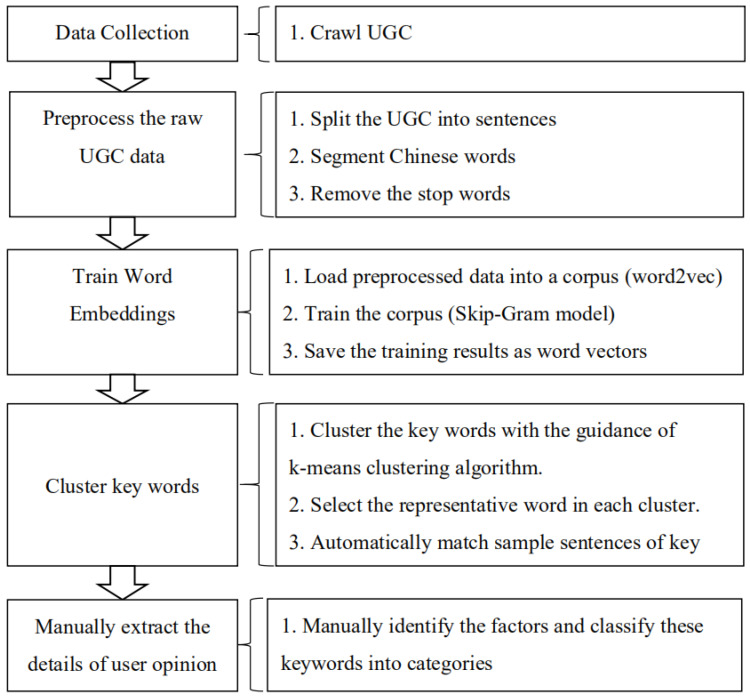
The methodological framework.

**Figure 4 ijerph-17-03905-f004:**
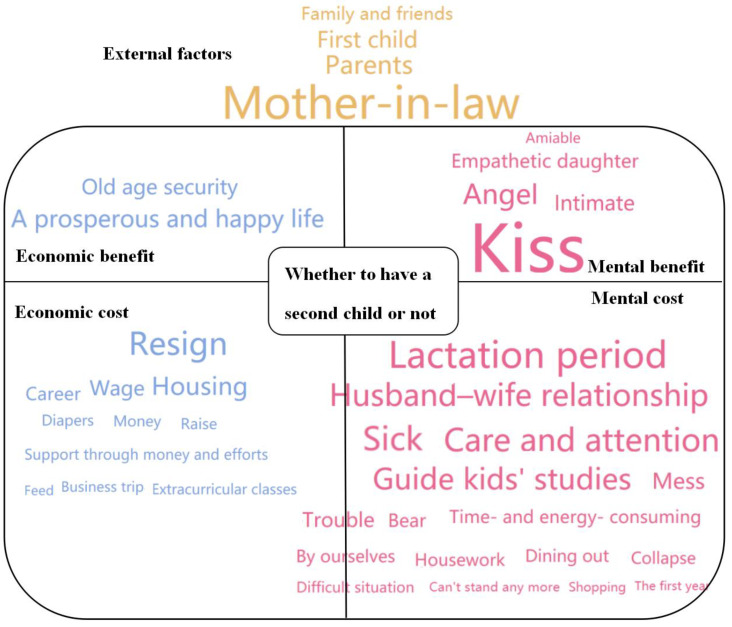
In-depth reasons for deciding whether or not to have a second child.

**Figure 5 ijerph-17-03905-f005:**
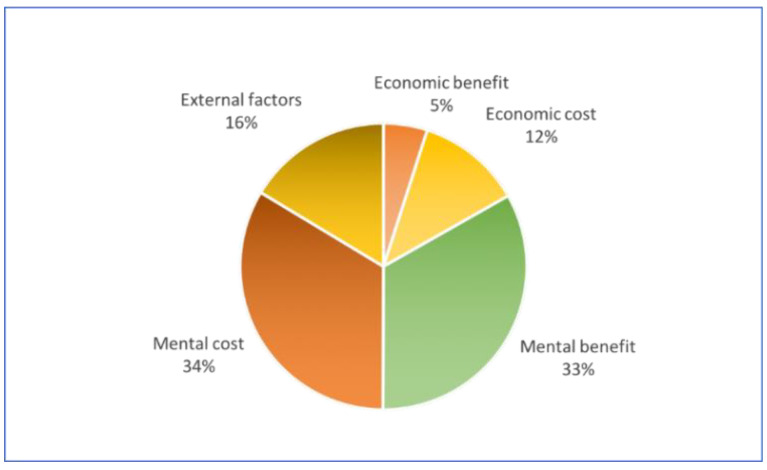
The frequency of keywords mentioned for each category.

**Table 1 ijerph-17-03905-t001:** The raw and processed data.

Raw Data	Processed Data
The polarization of wealth caused by rapid development has resulted in most households being unable to bear the cost of child-rearing.	polarization, wealth, rapid development, households, unable to, bear, cost of child-rearing
The first child is not brought up properly, so some people want to have a second child. However, they cannot sustain the vigor required to have a second child.	first, child, brought up, properly, second child, sustain the vigor
Having a second child will diminish the quality of my life. When it comes to parenting, there are significant differences between how we do it today and how it was done in the past. Nowadays, many parents send their children for extracurricular classes. No one is willing to let their children lag behind other children.	second child, diminish, quality, my life, significant, differences, parenting, between how we do it today, how it was done in the past, parents, send, children extracurricular classes, lag behind

**Table 2 ijerph-17-03905-t002:** Keywords indicating economic benefits and sentence examples.

Keywords	Count *	% *	Examples of Sentences
A prosperous and happy life	50	4.7%	Everyone tells me that with more children I will have a prosperous and happy life in the future. But I feel that parents will always be devoted to their children.
Old age security	39	3.9%	Even if you have children, will they provide you with old age security? The better the social security for the elderly is, the less willing people will be to have children, leading to a shrinking population.

***** Count is the number of times the keyword appears in the text and % represents the percentage of users who mentioned the keyword, which applies to Table 2, Table 3, Table 4, Table 5 and Table 6.

**Table 3 ijerph-17-03905-t003:** Keywords indicating economic expenditure and sentence examples.

Keywords		Count	%	Examples of Sentences
Short-term expense	Raise	15	0.9%	The extra money each month is used entirely for raising children.
	Money	14	1.4%	After we had a second child, my husband has been trying to make as much money as he can.
	Diapers	14	1.1%	After the delivery, a lot of money was spent on items for the baby, such as milk powder, diapers, and all kinds of daily necessities and toys.
	Support through money and efforts	13	1.3%	All of my classmates and other people I know who had a second child are those whose parents and parents-in-law can provide support through money and efforts to help them to raise the child.
	Business trip	11	1.0%	I travel a lot and have little time for kids. To be around with the first child, I tried to reduce social activities. There is no time for a second child.
	Feed	8	0.8%	To feed the baby, money is needed.
Long-term expense	Housing	55	5.5%	As planned, we had bought an apartment before we had the baby.
	Wage	37	3.1%	Regardless of the time and effort invested, the cost of extracurricular classes for the two children has already exceeded the average wage in Shanghai.
	Extracurricular classes	12	1.2%	How much does it take to raise a college-going child? Let’s just talk about education. The nine-year compulsory education is tuition-free, but the costs for extracurricular classes are high.
Indirect expense	Resign	83	8.4%	Left with no choice, I resigned to take care of my two children full-time.
	Career	28	2.8%	With a second child, can mothers devote themselves to their careers?

**Table 4 ijerph-17-03905-t004:** Keywords indicating mental benefits and sentence examples.

Keywords	Count	%	Examples of Sentences
Kiss	219	16.3%	Until the age of six, my child kept saying ‘I love you, mom,’ ‘kiss me, mom,’ and “’I will protect you, mom.’ Those were happy moments, when both of them kissed each other and cuddled together.
Angel	65	5.4%	The first child is an angel baby and is very sweet and considerate, so I was optimistic and had positive expectations regarding the second child. Both of my children are angels. They hardly cry and just groan a little when they feel sick or uncomfortable.
Intimate	37	2.4%	As soon as I get home from work, my son will cling to me. It is our intimate time.
Empathetic daughter	36	3.0%	Although a little tired, I still feel happy. My two little empathetic daughters bring me much joy.
Amiable	11	1.1%	I am very happy to see that my two children are so amiable and considerate and that they are very close to each other. Isn’t this the original motivation for having a second child?

**Table 5 ijerph-17-03905-t005:** Keywords indicating mental costs and sentence examples.

Keywords	Count	%	Examples of Sentences
Lactation period	131	11.4%	I am now in the lactation phase. I feel very tired taking care of the child every day. I don’t have any time left for myself at all.
Husband–wife relationship	101	4.1%	Before the first child was born, my relationship with my husband was very good. But after having the first child, we quarreled quite frequently. After the birth of the second child, we didn’t even quarrel anymore, as we had become numb. Our relationship has never recovered.
Sick	92	5.2%	If two children get sick at the same time, I want to collapse and die. I am too tired to even open my eyes in such a situation.
Care and attention	88	8.8%	Two children need nonstop care and attention. I think it is actually better to have only one child.
Guide kids’ studies	79	7.9%	I have to guide my kids’ studies, taking the first child for extracurricular classes, and checking and helping the homework.
Mess	47	4.7%	You should be mentally strong enough to face the mess in your life.
Trouble	39	3.9%	From the painful delivery until the second birthday of the second child, each and every day, I regretted having made so much trouble for myself.
Bear	31	3.1%	The first year and a half after delivery, and especially the first year, was too much to bear. I was already depressed.
Time- and energy-consuming	29	2.9%	It is time- and -energy-consuming to raise a child. There is no doubt that having two children is harder than having one.
By ourselves	26	2.3%	My husband and I are engaged in business. We raised the child by ourselves. I was exhausted during the first and second year after the baby was born.
Housework	26	2.1%	She does the housework and helps the child with homework. She never goes to bed early. We work day and night and are too tired and too sleepy.
Dining out	25	2.5%	Dining out with the two children means that you will never be able have a good meal, as one is crying and the other is running around. You will only be able to look at the dishes on the table.
Collapse	23	2.3%	I am so exhausted that I’m on the verge of collapse.
Difficult situation	16	1.6%	We are in a difficult situation that has gradually gotten worse.
Can’t stand any more	9	0.9%	I really can’t stand any more! Not another day like this! Apart from requiring money, it is time-consuming and energy- consuming.
Shopping	7	0.7%	There is no freedom for us now that we have two children. I never think about going out and doing some shopping.
The first year	6	0.6%	In the first year, life was very tough. After my maternity leave ended, I started to work and take care of the children at the same time.

**Table 6 ijerph-17-03905-t006:** Keywords indicating external factors and sentence examples.

Keywords	Count	%	Examples of Sentences
Mother-in-law	148	7.8%	The first child was raised by my mother-in-law. I was at ease and there was nothing to worry about! I have many petty conflicts with my mother-in-law, but because she took care of the baby, I just have to endure them all!
Parents	65	3.5%	My parents also wish that I would have two children. My parents pushed me to have a second child, but I didn’t want to. It’s too tiring.
First child	52	1.9%	Gradually, over time, when my first child reached the age of five years, he stopped objecting so much to us having a second child.
Family and friends	31	3.1%	The older generation of people, such as my parents-in-law, were constantly urging me to have a second child. Most members of my family and friends have given birth to two children. I followed them in having a second child.

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
