# Peer review of "Investigating Fertility Intentions for a Second Child in Contemporary China Based on User-Generated Content"

_ijerph, 2020, doi:10.3390/ijerph17113905_

Round 1

Reviewer 1 Report

I think the authors have made a good effort to address my recommendations

Reviewer 2 Report

The authors have applied all suggested changes in the manuscript. 

This manuscript is a resubmission of an earlier submission. The following is a list of the peer review reports and author responses from that submission.

Round 1

Reviewer 1 Report

The aim of the study is to "systematically investigate the micro-level factors 72 influencing the willingness of Chinese couples to have two children". The authors use a combination of natural language processing and manual coding, using data from an online Q&A site.

The main strength of the paper is the methodological approach, which is interesting and innovative. I think the findings are only partially able to address the research question, for reasons I explain below.

Overall, I suggest the authors reframe their paper as a methodological piece, discussing the strengths and weaknesses of their approach in answering social science questions. I provide some more suggestions below, separated by section.

  • Data is derived from answers to two questions on an online discussion platform. It is important to assess what these questions were really assessing.
    • The first question "“Why Chinese government’s policies encouraging a second child have limited effect?” refers to people's impression about the effect of the policy change on fertility behaviour in the general population, it does not refer to their own reasons or motivations for having or not having 2 children.
    • The question “Do you feel happy about having two children?” does refer to individual-level factors, but is unclear if the question applies only to those who already have two children, or whether it is asked in a hypothetical way. The authors should clarify this.
  • Sample selection: as responses were collected from an online forum, they do not constitute a representative sample. For example, internet users are likely to be more educated, younger etc. Moreover, people probably contributed to the forum because they had strong views about the topic, either positive or negative. Their responses are thus not representative for the general population and this should be made clear.
  • The limitations of the method need to be discussed in more detail. For example, while the method is able to extract keywords, it is not able to assess whether these were used in a negative way (for example "old age security" can be used as an argument for and against having a second child, as shown in table 2). Moreover, the algorithm is unable to cluster keywords into meaningful groups, only humans can do this. A more extensive discussion of what NLP can do and where human input is needed would be helpful, also for future research.
  • Other than counting keywords, the method is not able to rank the relative importance / prevalence of particular themes and arguments, which would be helpful in answering the research question.
  • Figure 1 is difficult to interpret, better to show separate population structure graphs for each decade or eliminate altogether.
  • Please explain how to interpret Figure 4: what is the meaning of the size, color and position of the words?

Reviewer 2 Report

This is a very interesting study, which makes use of user-generated content technique to understand how individuals in China make decisions about having a second child. This is a welcome contribution to the literature on reproductive decision making in general and in the context of China in particular.

However, I have several comments/suggestions for improvement:

The introduction:

There are some sections in the paper that are missing references to their sources. For example: lines 27-29 and 38-39. In addition, there is no reference to where the figures cited in lines 291 and 292 are taken from.

In addition, I was quite surprised that in the discussion of fertility trends in China, there was no mention of the one child policy, or any other population control policies that were introduced in the past decades. Furthermore, I would suggest expanding a little bit about the unique context of childbearing preferences in China (as opposed to Western countries). For example, in lines 64-65 the authors mention family size preferences in China, but do not say how these figures compare to other countries (e.g. the high preference of two-child families looks quite similar to fertility preferences in many other Western societies). Also, in line 62, it is mentioned that “In China, childbearing attitudes and behaviors are rooted in Confucian philosophy”, but the authors do not really explain how Confucian philosophy might shape fertility.

Presentation and discussion of findings:

A more efficient presentation of the findings would be helpful here. For example, the authors mention that the results in Table 3 can be organized under three sub-categories (short-term expenditures, long-term expenditures and indirect expenditure), so why not presenting a table or a figure with this concise classification? Also, the authors mention that “short-term expenditure was not a major concern. Thus, while keywords such as “diaper,” “feed,” and “raise” were mentioned, they were not emphasized.” (lines 205-206). However, it is unclear what the authors mean by that; were these factors mentioned less frequently than other costs? Overall, showing the proportions of respondents who gave different reasons in favour or against having a second child could help better understand the results of the survey (e.g. what is the proportion of respondents who emphasized benefits over costs of having a second child, or vice versa?)

Finally, it is worthwhile including future research directions, for example, it would be helpful if using the UGC would enable researchers to obtain some demographic information about the respondents, such as, age, gender, socioeconomic status, marital and family status etc.

By addressing these issues, I believe this manuscript could make a worthy contribution to the literature on fertility patterns in China.  

Round 2

Reviewer 2 Report

The authors have addressed most of the comments made on the previous version. However, there are a few more points that should be addressed in order to further enhance the quality of this manuscript:

The presentation of findings should be done more systematically to highlight the relative importance of benefits versus costs of having a second child. For example, in Figure 4, the size of the different key words should reflect the proportion of respondents who mentioned them as a share of the whole sample. This way, the word “Kiss”, which was mentioned by 22% of respondents (according to Table 4), should be much bigger than the key words related to “A prosperous and happy life”, which were mentioned by only 5% of respondents. This is just one example, but the whole figure should show the relative frequency of all key words among all respondents. Also, adding a figure that shows the proportion of key words mentioned for each category (i.e., economic benefits, economic costs, mental benefits, mental costs and external factors) would be helpful, as this will highlight that the most common key words are associated with the mental costs of having a second child.

As most previous studies about fertility preferences focus mainly on economic costs of childbearing, the finding that mental costs is by far the most common aspect mentioned by respondents is an important contribution of this study and should be highlighted (particularly in the discussion section).

Further minor comments:

While it is good that the authors present the proportions of each key word, the frequencies should also be presented in parentheses. The proportions should also be presented in a more consistent manner, i.e., using the same number of digits after the decimal point throughout the manuscript (preferably only one digit after the decimal point should be used).

In lines 315-318, the authors cite findings from a survey by the China Psychiatrists Association, but without a proper reference to this study.
